# Electrical Resistance of Stainless Steel/Polyester Blended Knitted Fabrics for Application to Measure Sweat Quantity

**DOI:** 10.3390/polym13071015

**Published:** 2021-03-25

**Authors:** Qing Chen, Lin Shu, Bailu Fu, Rong Zheng, Jintu Fan

**Affiliations:** 1Shanghai International Fashion Innovation Center, Donghua University, Shanghai 200051, China; chenqing@dhu.edu.cn (Q.C.); rzheng@dhu.edu.cn (R.Z.); 2School of Electronic and Information Engineering, South China University of Technology, Guangzhou 510640, China; 3The Institute of Modern Industrial Technology of SCUT in Zhongshan, Zhongshan 528400, China; 4The Institute of Textiles and Clothing, The HongKong Polytechnic University, Hongkong, China; jin-tu.fan@polyu.edu.hk

**Keywords:** knit, electrical resistance, liquid sweat, hydrophilicity

## Abstract

Skin wetness and body water loss are important indexes to reflect the heat strain of the human body. According to ISO 7933 2004, the skin wetness and sweat rate are calculated by the evaporative heat flow and the maximum evaporative heat flow in the skin surface, etc. This work proposes the soft textile-based sensor, which was knitted by stainless steel/polyester blended yarn on the flat knitting machine. It investigated the relationship between electrical resistance in the weft/warp directions and different water absorption ratio (0–70%), different sample size (2 cm × 2 cm, 2 cm × 4 cm, 2 cm × 6 cm and 2 cm × 8 cm). The hydrophilic treatment effectively improved the water absorption ratio increasing from 40% to 70%. The weft and warp direction exhibited different electrical behaviors when under dry and wet conditions. It suggested the weft direction of knitted fabrics was recommended for detecting the electrical resistance due to its stable sensitivity and linearity performance. It could be used as a flexible sensor integrated into a garment for measuring the skin wetness and sweat rate in the future instead of traditional measurements.

## 1. Introduction

Body fluid is important for human beings. In a different environment, it can reflect the health condition. For instance, sweat is a kind of moisture created by human beings. The skin wetness can reflect the body condition of different people, such as firefighters, babies, elderly people, soldiers, sportsmen, etc. Smart textiles have emerged from the combination of textiles and electronics [1]. The potential market is wide, e.g., for sports, healthcare or military [1].

Previously, the measurement of skin sweat volume included qualitative and quantitative measurements. For the quantitative method, three are the nude body weight difference method, dress weight difference method, clothing weight gain method for measuring or calculating the whole body’s sweat volume. For determining the local body sweat volume, the normal method is to use the dry material with better absorption ability attached to the parts of the body and to weigh the difference of the material after sweating. However, those methods are not convenient to operate, not accurate especially for calculation method. Moreover, it cannot detect the time of starting sweat and the sweat rate during the whole process.

With the increasing interest in wearable electronic systems, new conductive materials have been developed for sensing, actuating and signal transmission. Conductive components (metal, carbon or metal salt particles) can be added to the textiles in all stages of the production process (fiber, yarn and fabric formation, coating, and embroidery) using conventional or new techniques. Many countries have become an aging society, which will increasingly require domestic healthcare systems to provide elders long-term health monitoring [2]. Wearable sensors have the potential to provide new methods of non-invasive physiological measurement in real time. Textile-based humidity sensors could be an important component of smart wearable electronic textiles and have potential applications in the management of wounds, bed-wetting, and skin pathologies or for microclimate control in clothing [3], biomedical or domestic applications [4], humidity level examination building walls [5], carpets and mattresses [6], automotive applications [7], smart wound dressings and bandages [8,9].

To date, several approaches to transfer conventional capacitive, impeditive, and resistive humidity sensors onto textiles have been developed. Temperature and humidity sensors are often solved as rigid sensors on large-area flexible substrates [10,11], which are stitched to a textile or as a sensor system on flexible polyimide substrates, which are woven into a textile [12,13] or as printed sensors directly onto the surface of textiles by the use of screen printing technology [1,14]. Embroidered temperature and humidity sensor elements for smart textile applications in health and medical care were presented [2]. A textile-based humidity sensor was demonstrated using conductive yarns woven into textiles as electrodes and cotton as a hygroscopic material [15]. Electrodes were printed, and a sorption layer was deposited [16]. Printed humidity sensors on a polymer tape and subsequently woven the sensors into a textile structure [2,13]. Sensor electrodes were printed on textile with Ag nanoparticles ink using ink-jet technology [2]. Carbon nanotubes were deposited onto glass fiber surfaces to detect humidity [17]. A thin carbon nanotube film was used onto cotton threads to measure humidity and albumin in blood [18]. A moisture monitoring system with textile-integrated sensors for wound healing assessment [19].

Theoretically, textile humidity sensors may be divided into passive and active humidity sensors. In the structure of a passive sensor, moisture sensing material is a textile fabric. Such textile sensors are able to efficiently react only one-way, i.e., to detect the presence of moisture or the increase of humidity in the environment. Hence, applications of such sensors are applied to geotextiles and monitoring in sportswear when the sweating rate is high or skin resistance response is very good [20]. The operation principle of resistive humidity sensors is based on measuring the changes in electrical impedance in the hydroscopic medium. Sensors usually consist of precious metal electrodes deposited on a substrate coated with a hygroscopic material, such as a conductive polymer, salt or other activating chemicals [21]. The hygroscopic material absorbs water, and ionic functional groups are dissociated, resulting in an increase in conductivity. Thus, as the humidity increases, the resistance of the material decreases. Resistive sensors show an exponential response to the humidity changes, which is linearized by analog and digital methods [22].

Although these approaches represent significant progress toward fabricating textile-based humidity sensors, major challenges remain to be overcome [19]. For example, the durability of electrodes prepared by printing, deposition, and coating remains an important issue to be addressed. Damage on the surface of the coated conductive layer may decrease the sensitivity of the sensor. These solutions are their poor washing resistance and usually only point measurement. It is also uncomfortable when deposited film makes intimate contact with human skins for a long time. Moreover, the poor stretchability of glass fibers deposited by conductive layers cannot be stretched to accommodate the movement of the subject. Although it has provided a promising route to realize flexible sensing arrays, the applications of the flexible sensors are limited to nearly flat substrates.

Shou et al. studied the mass and heat transfer property and mechanism of fiber assembly to develop high-performance moisture transport and thermal protective garment [23,24,25,26,27]. Hence, it is important to evaluate the moisture transport property of newly developed products. This project conducts a systematical investigation into a highly sensitive network with flexible and stretchable humidity sensors integrated on a permeable, elastic, thin, and lightweight knitted garment. The fabric sensing network was achieved by physically linking distributed fabric sensors with stretchable knitted interconnects. The textile-based conductive knitted fabric can be used as an effective textile-based sensor with different content and fiber blended. It can be used to directly measure the skin wetness and sweat rate instead of traditional calculation based on evaporative heat flow in the skin surface [28].

## 2. Materials and Methods

### 2.1. Materials

Stainless steel fiber and polyester fiber blended conductive yarn was selected (see Figure 1). The yarn count was 30 s/2, in which the polyester fiber was 80%, and stainless steel fiber was 20%. The conductivity of conductive yarn blended with stainless steel fibers and polyester fibers is 3.48 × 106 Ω/cm. 1 + 1 rib knitted fabric was knitted by a flat knitting machine (Stoll, Neufra, Germany, machine gauge E14).

The knitted fabrics were treated with a hydrophilic softener in order to increase the water absorption capacity. The fabrics were immersed in a 20% hydrophilic finishing agent and 80% water for 15 min at 25 °C and then dried under 160 °C for 2 min. The knitted fabrics with non-finishing and hydrophilic finishing are shown in Figure 2a,b show, and Table 1 is the basic fabric parameters. The contact angle measured under 3 uL drop by DCAT-21 (Dataphysics Instruments, Stuttgart, Germany). The results are shown in Figure 3a,b.

### 2.2. Methods

The fabrics are cut along the transverse and longitudinal directions in size of 2 cm × 2 cm, 2 cm × 4 cm, 2 cm × 6 cm and 2 cm × 8 cm. The electrical resistance is measured by multimeter (Tektronix, Beaverton, OR, USA). The samples are measured in the dry state and in a different wet state (the water absorption of fabric is from 0–70%). Physiological saline solution (concentration of NaCl is 0.9%) was dropped onto the conductive knitted fabrics. Water absorption ratio, water absorption weight, and sensitivity were calculated as below.
(1)$$\mathrm{Water}\;\mathrm{absorption}\;\left(\%\right)=\frac{{\mathrm{weight}\;}_{\mathrm{wet}\;\mathrm{fabric}}-{\mathrm{weight}\;}_{\mathrm{dry}\;\mathrm{fabric}}}{{\mathrm{weight}\;}_{\mathrm{dry}\;\mathrm{fabric}}}\times 100\%$$
(2)Δlogr = logr(i) − logr(0)
(3)$$\mathrm{Water}\;\mathrm{absoption}\;\mathrm{weight}\;\left(g\right)={\mathrm{weight}\;}_{\mathrm{wet}\;\mathrm{fabric}}-{\mathrm{weight}\;}_{\mathrm{dry}\;\mathrm{fabric}}$$

The sensitivity *α* is defined by:(4)$$\alpha (\%)=\frac{\log r(WAR\max)-\log r(WAR\min)}{WAR\max-WAR\min}\times 100$$

*WAR*max is the maximum water absorption ratio. *WAR*min is the minimum water absorption ratio [29].

## 3. Results and Discussion

The treated fabrics had 33% higher mass and 40% lower thickness than untreated fabrics. This is because the fabric shrinkage happened during the wetted finishing process. As a result, the treated fabric had more loop numbers in a unit area. After hydrophilic finishing, the hydrophilicity was improved. The contact angle of untreated and treated fabrics was 120.2° and 99.4°, respectively.

### 3.1. The Effect of Water Content on the Electrical Resistance of Untreated and Treated Knitted Fabrics

Since the electrical resistance of untreated fabric under dry conditions ranges from 0.25 to 3.2 MΩ in the weft direction and from 22.57 to 100.75 MΩ in the warp direction with the length at 2 cm, 4 cm, 6 cm, and 8 cm and width at 2 cm. However, the electrical resistance of fabric under 10–40% water absorption condition ranged from 34.92 to 2101 kΩ for the weft direction and from 35.98 to 6875 kΩ for the warp direction.

Regarding treated fabrics, the electrical resistance of fabric under dry conditions ranges from 3.02 to 39.67 MΩ in the weft direction and from 43.75 to 210.75 MΩ in the warp direction with the length at 2 cm, 4 cm, 6 cm, and 8 cm and width at 2 cm. However, the electrical resistance of fabric under 10–70% water absorption condition ranged from 6.27 to 4115 kΩ for the weft direction and from 6.91 to 2911 kΩ for the warp direction. In order to compare the electrical resistance under dry and wet conditions, all values were converting into a logarithm.

As Figure 4a–d shows, the highest water absorption ratio detected by untreated fabric was 40%, while that was 70% for treated fabrics. This is because the fabric cannot absorb more water to this point, which was determined by the hydrophilicity and water absorption ability of the fabric.

Evidently, the electrical resistance of knitted fabrics decreased gradually with the increase of water content (Figure 4). It was decreased in the weft direction from 2.4 to 1.54 for the untreated 2 × 2 sample, from 2.95 to 1.96 for the untreated 2 × 4 sample, from 3.33 to 1.81 for the untreated 2 × 6 sample, from 3.51 to 2.03 for the untreated 2 × 8 sample. In addition, it was decreased in the warp direction from 4.35 to 1.56, from 4.61 to 1.77, from 4.85 to 1.65, from 5 to 1.82 for the corresponding sample size. As for treated fabric, the values in the weft direction declined from 3.48 to 0.8 for the treated 2 × 2 sample, from 3.84 to 0.86 for the treated 2 × 4 sample, from 4.27 to 1.06 for the treated 2 × 6 sample, from 4.6 to 1.07 for the treated 2 × 8 sample; while in the warp direction, the values declined from 4.64 to 0.84, from 4.85 to 0.86, from 5.01 to 0.89, from 5.32 to 1.01 for the corresponding sample sizes. This is caused by the fact that the water molecules and NaCl in the material, which had lower conductivity than conductive fabric. The electrical resistance of fabrics within the same area decreases with the increase of moisture content. This is because the greater the moisture content, the larger the wetted volume. Therefore, the higher the moisture content, the better the conductivity of fabrics and the smaller the measured resistance value. It was found that the electrical resistance of treated fabric in dry conditions was higher than that of untreated fabric. This is due to the fabric after treatment; there was a shrinkage, which increased the fabric mass and thickness.

The difference among untreated and treated samples 2 × 2, 2 × 4, 2 × 6 and 2 × 8 under 0%, 10%, 20%, 30%, 40%, 50%, 60% and 70% water absorption condition were examined. The values of *p* are less than 0.05, so there were significant differences (see Table 2 and Table 3).

Figure 4a,b plots log (electrical resistance) values as a function of water absorption for untreated fabrics. The nonlinear regression test of electrical resistance and water absorption ratio was processed by regression-curve estimation (SPSS). Regarding warp direction, the regression equation between water absorption and log *r* is y = 4.264 × 2.718^−0.025x^ with adjusted R^2^ at 0.804, *p* = 0.000. In the weft direction, y = 3.154 × 2.718^−0.013x^, adjusted R^2^ = 0.589, *p* = 0.000. Notably, highly regression between water absorption ratio and electrical resistance for treated fabric (Figure 4c,d). Regarding warp direction, the regression equation between water absorption and log *r* is y = 4.079 × 2.718^−0.023x^ with adjusted R^2^ at 0.929, *p* = 0.000. In the weft direction, y = 4.120 × 2.718^−0.021x^, adjusted R^2^ = 0.926, *p* = 0.000.

Water absorption weights of untreated and treated fabrics were drawn in Figure 5a,b. The water absorption weight increased with the fabric weight and size. However, treated fabrics could absorb more water due to their lower contact angle and higher mass when compared with untreated fabrics. Regarding untreated fabrics (Figure 5a), the larger sample size had higher water absorption capacity. The maximum water capacities of untreated fabrics were approximately 0.0328, 0.0656, 0.0984 and 0.1312 g for 2 × 2, 2 × 4, 2 × 6 and 2 × 8, respectively. Meanwhile, the maximum water capacities of treated fabrics were approximately 0.0764, 0.1529, 0.2993 and 0.3057 g for 2 × 2, 2 × 4, 2 × 6 and 2 × 8, respectively (Figure 5b). The maximum increases reached 133.04% when comparing the untreated and treated fabric. Moreover, the water absorption weight linearly increased with increasing water absorption ratio.

When comparing the values under each wet condition with the dry condition (∆log *r*/log *r*_0_ × 100%), there was a decline for all 4 sample sizes. As Figure 6a,b shows, the curves were not smooth for untreated fabrics. However, after the treatment, as Figure 6c,d shown, it can demonstrate the good linear relationship between water absorption and electrical resistance. The decreases in electrical resistance of treated fabric under different water absorption ratios were from 17.84% to 77.49% in the weft direction and from 34.91% to 82.19% in the warp direction. The decline of electrical resistance in the weft direction for the treated fabric was more stable than that in the warp direction.

Hence, the regression model was established by regression-curve estimation (SPSS) as below. “*y*” is the decreasing percentage of electrical resistance in the weft direction for treated fabrics, which was determined by water absorption ratio (*x*).
(5)$$y=-1.159-1.813\times x+0.016\times x^{2}-7.674\times 10^{-5}\times x^{3}\;(\mathrm{adjusted}\;R^{2}\;=\;0.993)$$

### 3.2. The Effect of Sample Size on the Electrical Resistance of Untreated and Treated Knitted Fabrics

As Figure 7a–d shows, the value of resistance is positively correlated with the length of the fabrics when the width of the fabrics remains at 2 cm. This is because the electrical resistance of knitted fabrics included the length-related resistance and contact- resistance. According to the equation of resistance, R = ρL/S, ρ is conductivity, L is the length, S is the area. When the length of fabric is enhanced, the total resistance is enhanced correspondingly.

Obviously, the electrical resistance is linearly correlated with the sample size for fabric after hydrophilic treatment. Since the treated fabrics could provide detecting values more stably, it was found that the slope of different curves between different treated sample sizes becomes gradually smaller from dry condition to higher water content. With the increase of water absorption ratio, the difference between samples reduced. For instance, in the warp direction of treated fabric, for sample 2 × 2, the differences were 2.33 between 0% and 10%, 0.35 between 10% and 20%, 0.31 between 20% and 30%, 0.26 between 30% and 40%, 0.25 between 40% and 50%, 0.14 between 50% and 60%, and 0.08 between 60% and 70%, respectively. However, for sample 2 × 8, the differences were 1.86 between 0% and 10%, 0.74 between 10% and 20%, 0.64 between 20% and 30%, 0.38 between 30% and 40%,0.31 between 40% and 50%, 0.2 between 50% and 60%,0.18 between 60% and 70%. It proved that in the initially wet condition, the total resistance of fabric was determined by the dry fabric, while in the highly wet condition, the total resistance of fabric was determined by physiological saline solution.

Figure 8a,b shows unstable increasing percentages for untreated fabrics, which had dramatic vibration in the weft or warp direction for several wet conditions. However, the treated fabrics could detect the variation of electrical resistance more stably (Figure 8c,d). When treated fabrics were under dry conditions, there were 10.22%, 22.7% and 32.14% increases in the weft direction when fabric length increased by 2 cm. The maximum increase percentage of “45.60%” in the weft direction was observed under 30% water absorption. The maximum increase percentage of “49.76%” in the warp direction occurred under 10% water absorption. For untreated fabrics, the maximum values were 61.54% in the weft direction and 71.94 in the warp direction. The increasing resistance percentage caused by the sample size was unstable when compared with that caused by water content. This may be that there were errors caused by hand cutting or contact point, while water content can be controlled accurately by micropipette. In conclusion, the conductivity of treated fabric was more stable than that of untreated fabric.

### 3.3. The Comparison of Warp and Weft Direction of Untreated and Treated Knitted Fabrics

The result of electrical resistance in the warp and weft direction untreated fabric was plotted in Figure 9a–d. The warp direction had higher electrical resistance than that of the weft direction in dry conditions. However, there existed a cross point between two curves for every sample size. The cross points were at 30% water absorption ratio for 2 × 2 cm sample, at 20% for both 2 × 4 cm sample and 2 × 6 cm sample, at 23% for 2 × 8 cm sample.

The results of the electrical resistance of treated fabrics were plotted in Figure 9e–h. The resistance in the warp direction is larger than that in the weft direction at the initial condition. It is consistent with the finding for untreated fabrics. For samples at 2 × 2, 2 × 4, 2 × 6, 2 × 8 (cm × cm) size, there were 33.36%, 26.34%, 17.31% and 15.74% increase, respectively. This is determined by the construction of conductive yarn interconnection in fabric. In the weft direction, there were more overlapped loops, which created more contact resistance in parallel connections. In the warp direction, the length-related resistance had a higher proportion than contact-resistance.

More interestingly, after treated fabric absorbing physiological saline solution, the electrical resistance in the weft direction was higher than that in the warp direction. This may be that when the fabric was wetted, the electrical resistance was determined by the resistance of water content within fabrics instead of the resistance of conductive fabrics.

With the water absorption ratio increasing, the resistance of difference decreased from 19. 1% to 5.25% for sample 2 × 2. However, for other sample sizes, first, there was an increasing difference between weft and warp direction at 30% or 40% water absorption ratio and then a decrease with the water content increase. With the length increasing, the resistance of difference becomes larger due to the larger length-related resistance.

### 3.4. The Comparison Sensitivity of Untreated and Treated Knitted Fabrics

The effect of sample size on the sensitivity is shown in Figure 10. As the sample size increased from 2 × 2 to 2 × 8, the sensitivity of untreated fabrics increased gradually from 5.43, 5.69, 5.88 and 6.16 (log *r*/% WAR) in the warp direction, 3.83, 4.25, 4.58 and 5.05 (log *r*/% WAR) in the weft direction. The treated fabric exhibited the highest sensitivity for each sample size; there were 6.22, 6.59, 6.93, and 7.15, respectively, when compared with the other three lines in Figure 10.

The treated fabric had lower sensitivity than the untreated fabric in the warp direction. This is because the untreated fabrics had a narrow sensing range only to 40% water absorption, which resulted in a lower value WARmax−WARmin. The warp direction had higher sensitivity than that in the weft direction. This may be that the maximum electrical resistance in the warp direction under dry conditions (0% water absorption ratio) was higher than that in the weft direction. It caused the differences between 0% and 70% were larger than the weft direction. However, for weft direction, the untreated and treated fabrics had a similar sensitivity. This proved that the sample size had impacted the sensitivity of both directions, while the treatment had an effect on the sensitivity in the warp direction but no effect in the weft direction.

### 3.5. The Statistical Model of Electrical Resistance Based on Sample Size, Water Absorption Ratio for Treated Fabric in the Weft Direction

According to the investigation above, the treated fabric in the weft direction had better electrical performance under different wet conditions. Hence, the statistical model was established in order to predict the electrical resistance by sample size and water absorption ratio. A very good quantitative agreement was found between the relative change of electrical resistance and the water absorption and sample size (adjusted R^2^ is 0.97888). Y represented the log (electrical resistance). The equation was plotted in Figure 11.
(6)$$Y=3.851-0.03942\times \mathrm{sample}\;\mathrm{size}-0.06627\times \mathrm{water}\;\mathrm{absorption}\;\mathrm{ratio}+0.00502\times {\mathrm{sample}\;\mathrm{size}}^{2}\;+3.60297\times 10^{-4}\times {\mathrm{water}\;\mathrm{absorption}\;\mathrm{ratio}}^{2}-3.21438\times 10^{-4}\times \mathrm{sample}\;\mathrm{size}\;\times \;\mathrm{water}\;\mathrm{absorption}\;\mathrm{ratio}$$

## 4. Conclusions

In summary, we fabricated conductive fabric, which can detect the change of water absorption ratio by measuring electrical resistance. Hydrophilic treatment can help to improve the water absorption property, linear regression between water absorption and electrical resistances, which determined a wider detecting range and more stable reading. Regarding treated fabrics, there were about 82.19% and 77.49% reductions under higher water absorption (70%) relative that under dry conditions (0%) in the warp direction and weft direction. It is observed that the values of log *r* in the weft direction were higher than that in the warp direction in the wetted condition, while there was an opposite observation under dry conditions. The larger size of fabric can absorb more water, but not a higher water absorption ratio. Measurement range is up to 0.3057 g water quantity for 2 × 8 sample size in this study. This smart knitted fabric will facilitate the design of a novel wearable humidity sensor to monitor blood leakage, human sweating, and underwear wetting in different body parts.

In the future, the sensitivity of the knitted fabric can be further improved by increasing the sensitivity of the metal material and hydrophilicity of yarn, which determines the sensitivity and the detecting range of water absorption ratio. The response time and reversibility of sensors are important parameters for practical applications that also should be investigated.

## Figures and Tables

**Figure 1 polymers-13-01015-f001:**
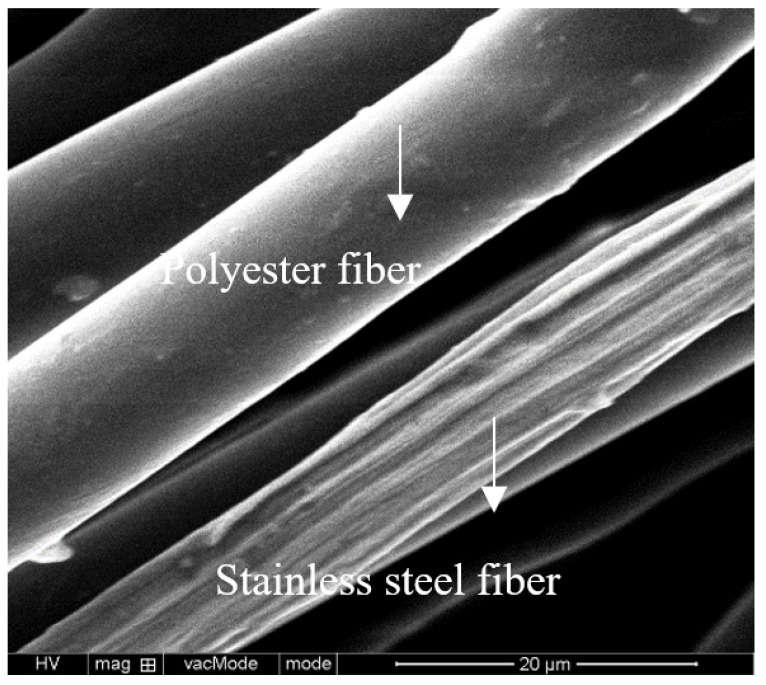
Stainless steel fiber and polyester fiber.

**Figure 2 polymers-13-01015-f002:**
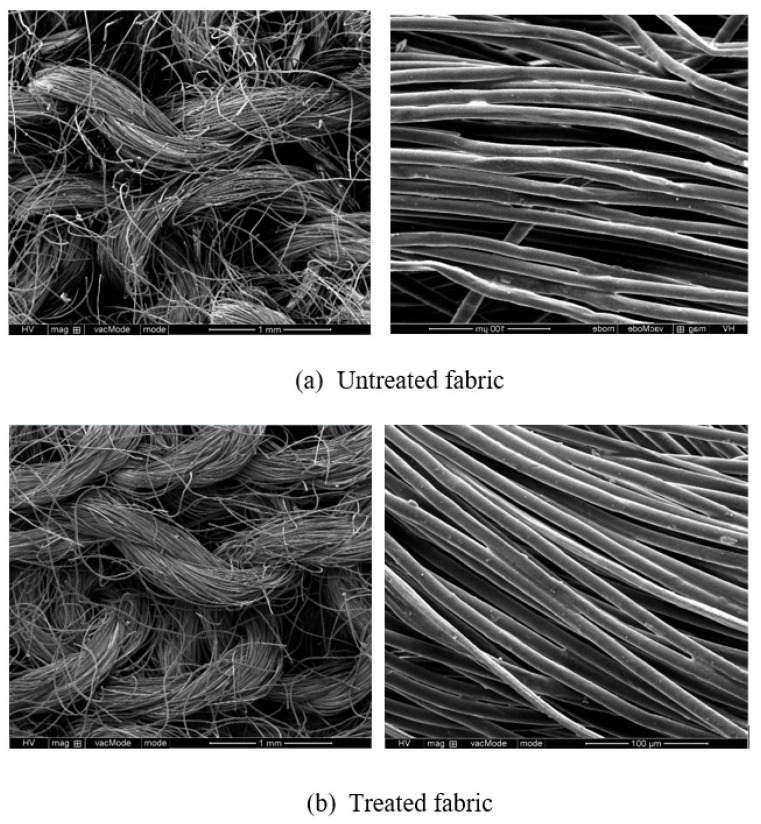
The knitted fabrics with non-finishing and hydrophilic finishing. (**a**) the surface of fabric and fibers are not smooth. (**b**) the surface of fabric and fibers are smooth.

**Figure 3 polymers-13-01015-f003:**
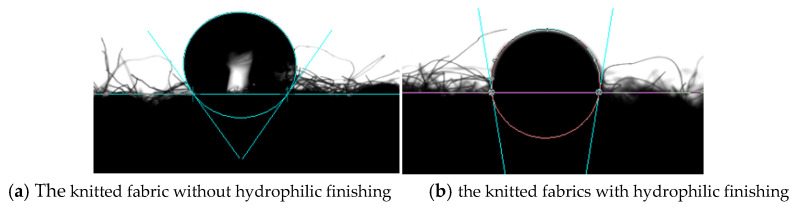
The contact angle of conductive knitted fabrics ((**a**) 120.2° for untreated fabric and (**b**) 99.4° for treated fabric).

**Figure 4 polymers-13-01015-f004:**
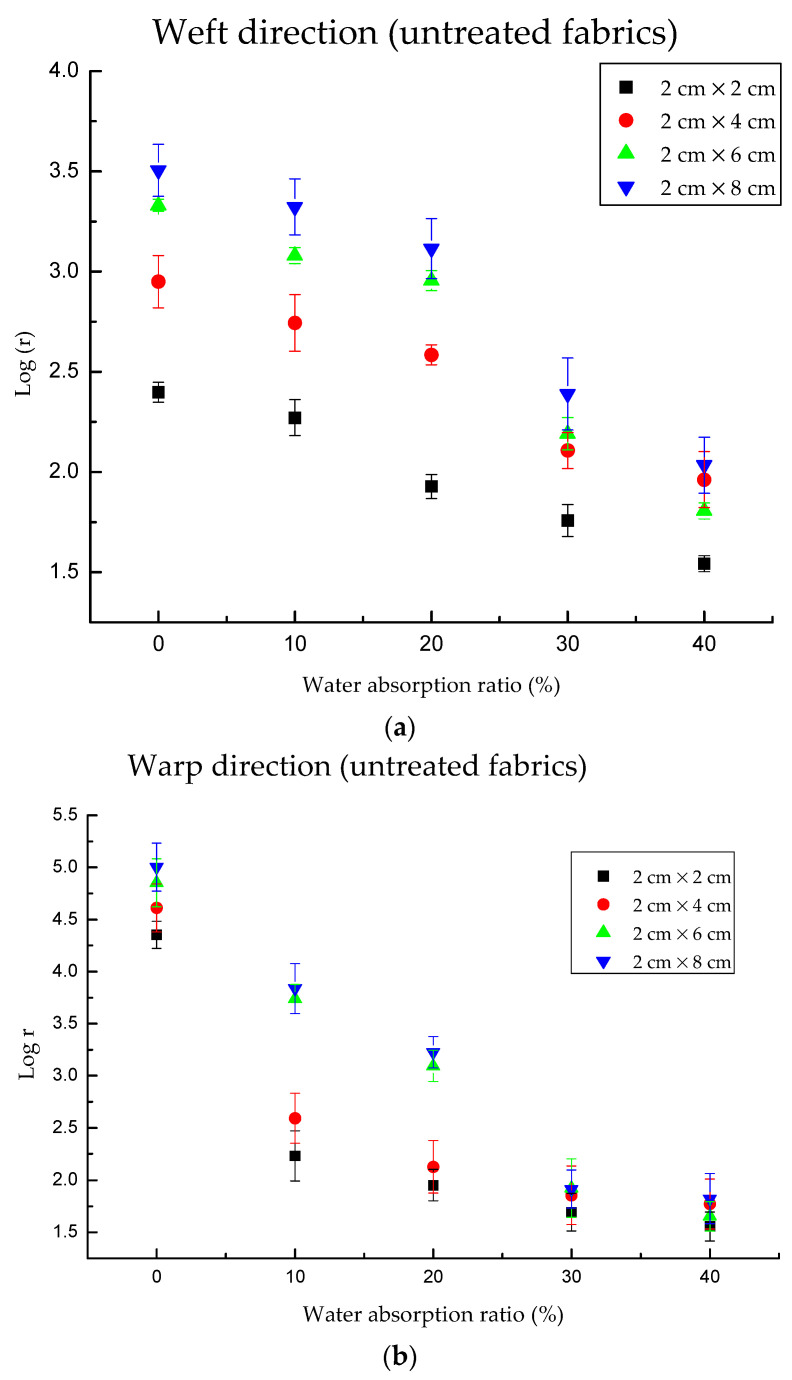
The electrical resistance of knitted fabric as a function of water absorption ratio. Note: y is log *r*, x is water absorption (%). (**a**) The electrical resistance in the weft direction for untreated fabrics; the maximum value of water absorption ratio is 40%, and the maximum of log *r* is about 1.5. (**b**) The electrical resistance in the warp direction for untreated fabrics; the minimum value of water absorption ratio is 40%, and the maximum of log *r* is about 1.8. (**c**) The electrical resistance in the weft direction for treated fabrics; the maximum value of water absorption ratio is 70%, and the maximum of log *r* is about 0.8. (**d**) The electrical resistance in the warp direction for treated fabrics; the maximum value of water absorption ratio is 70%, and the maximum of log *r* is about 0.84.

**Figure 5 polymers-13-01015-f005:**
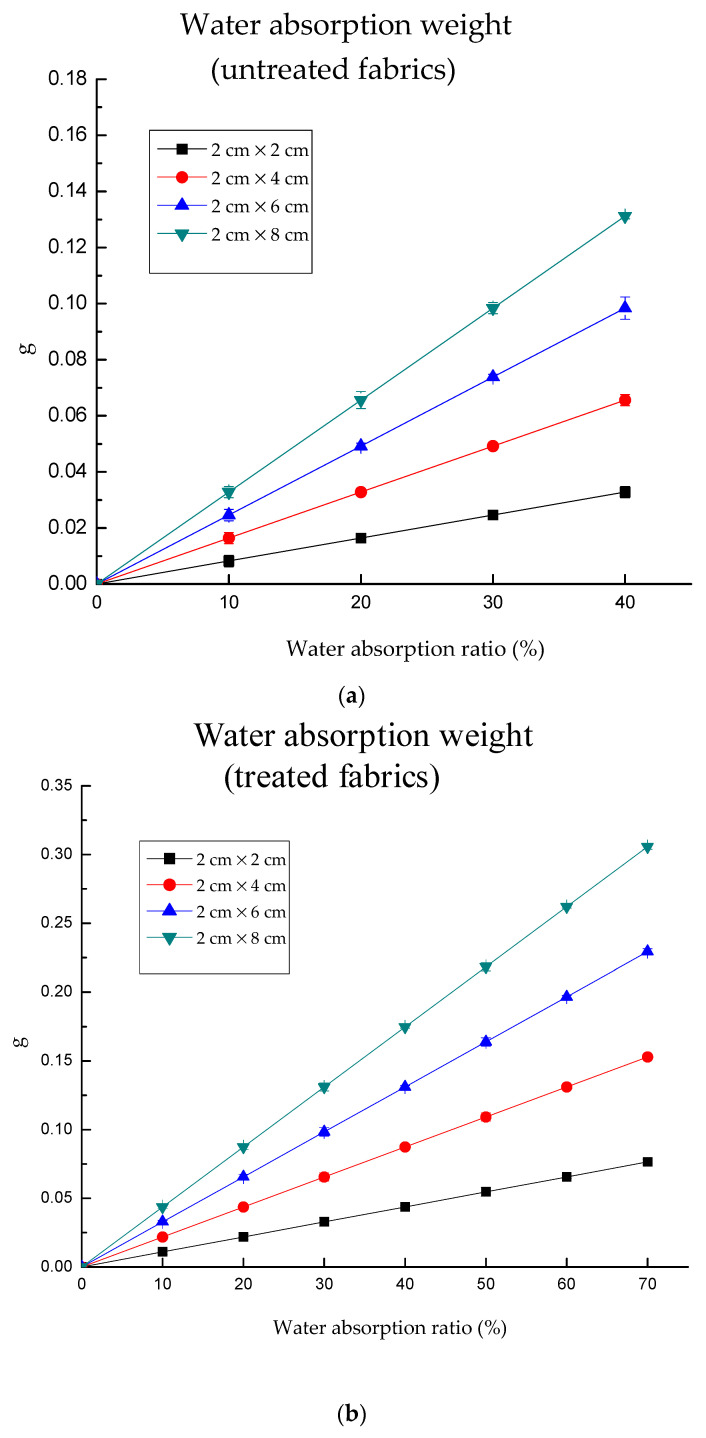
The water absorption weight of untreated and treated fabrics. (**a**) the maximum values of water absorption weight of the untreated fabric are about 0.13 g for 2 × 8 cm at 40% water absorption ratio; (**b**) the maximum values of water absorption weight of the untreated fabric are about 0.3 g for 2 × 8 cm at 70% water absorption ratio.

**Figure 6 polymers-13-01015-f006:**
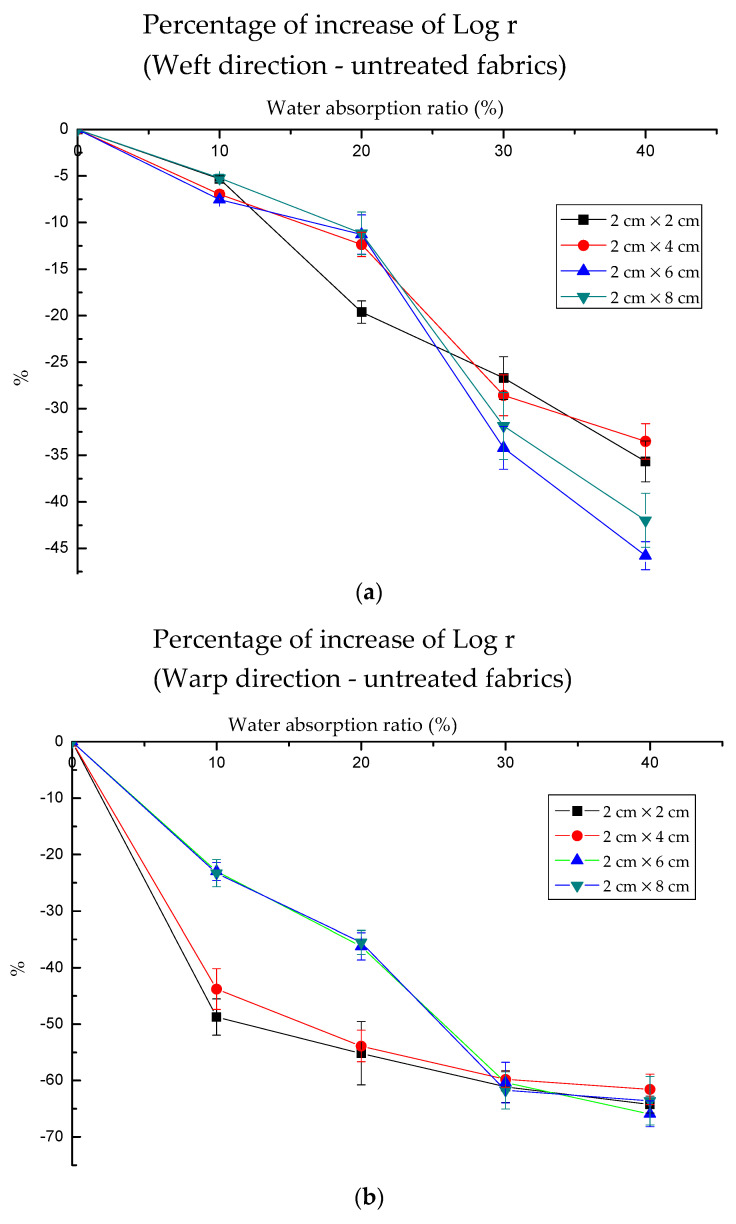
The percentage of the decline of log (electrical resistance) for each sample in the warp and weft directions when comparing the samples under the different water absorption. The declining trend of untreated fabrics is more stable when compared with that untreated fabric. (**a**) the percentage of log *r* for untreated fabric at weft direction; (**b**) the percentage of log *r* for untreated fabric at warp direction; (**c**) the percentage of log *r* for treated fabric at weft direction; (**d**) the percentage of log *r* for treated fabric at warp direction.

**Figure 7 polymers-13-01015-f007:**
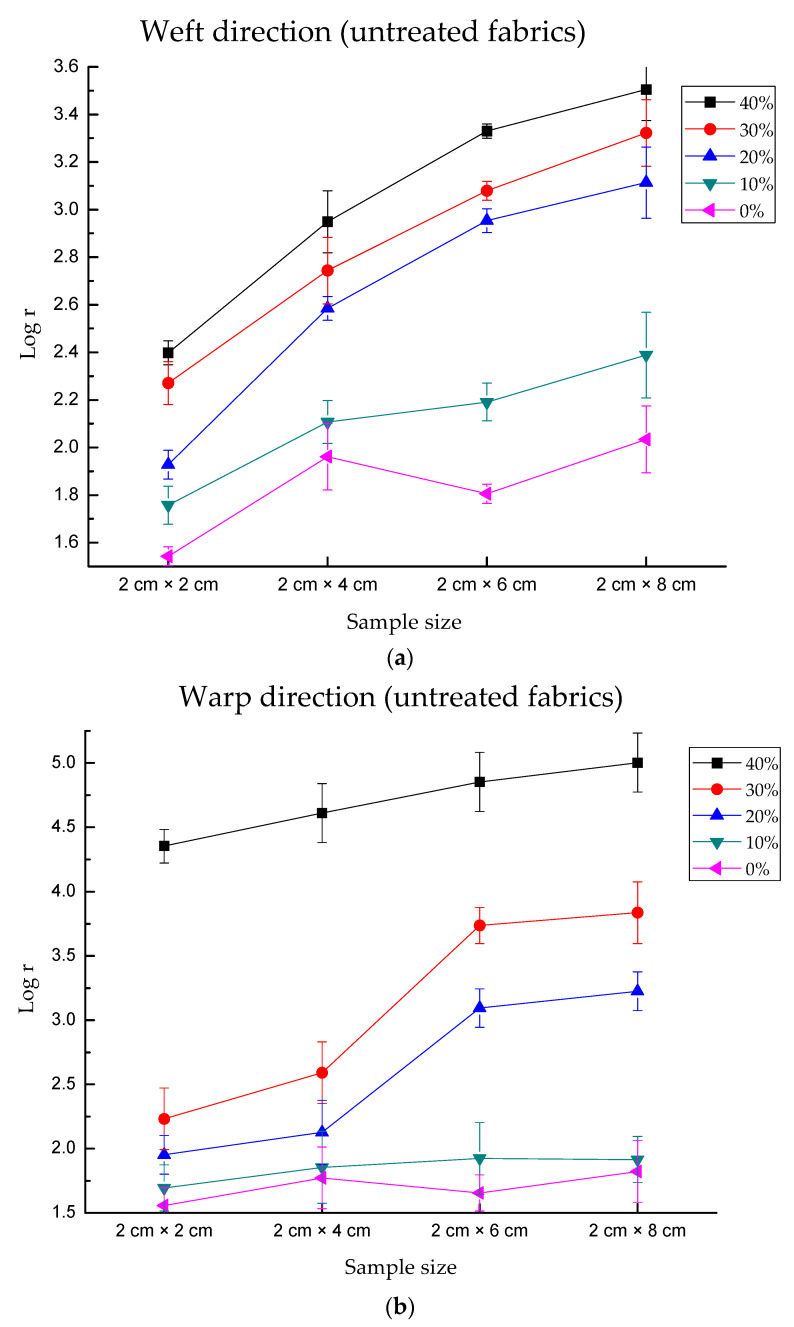
The results of electrical resistance of fabric as a function of sample size. It is found the weft direction of treated fabric had a more stable property.

**Figure 8 polymers-13-01015-f008:**
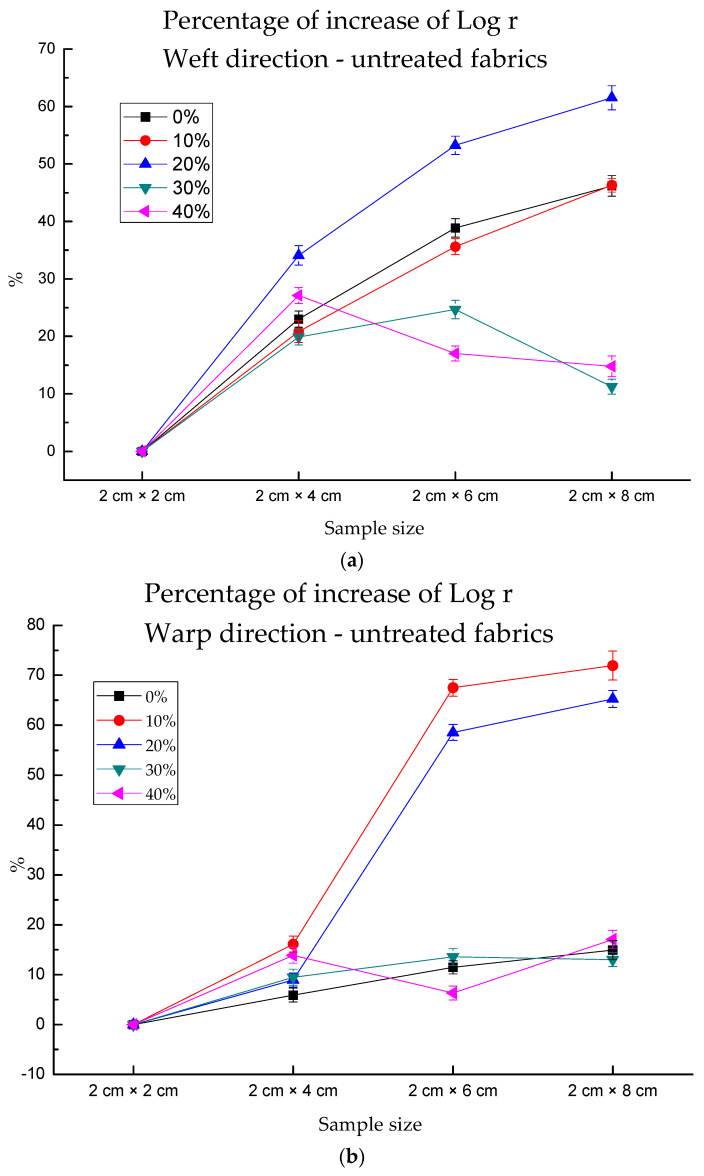
The increased percentage of log (electrical resistance) for each sample in the warp and weft directions when comparing the samples with different sizes under varying water absorption conditions.

**Figure 9 polymers-13-01015-f009:**
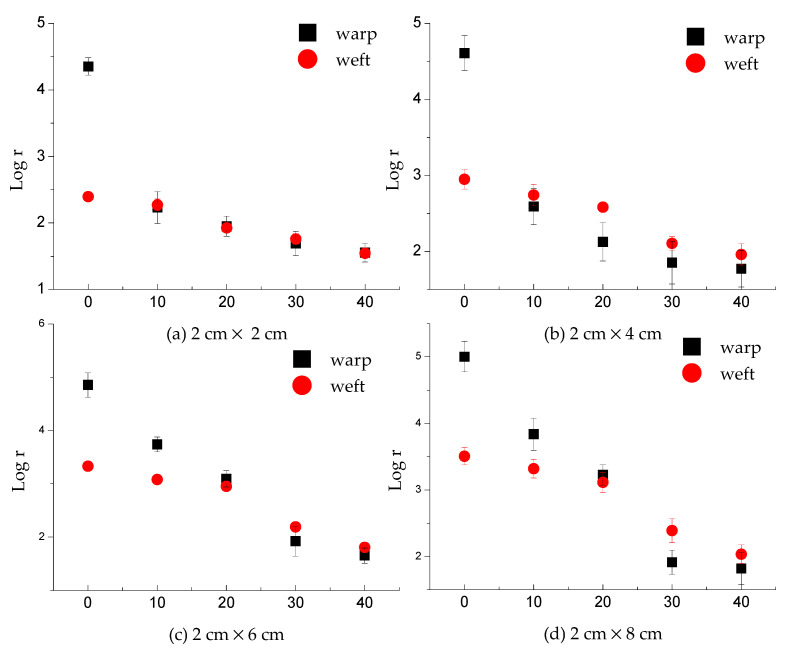
The comparison of log *r* of warp and weft directions. For treated fabric, under dry conditions, the values in the warp direction were higher than that in the weft direction, but under wet conditions, it was the inverse.

**Figure 10 polymers-13-01015-f010:**
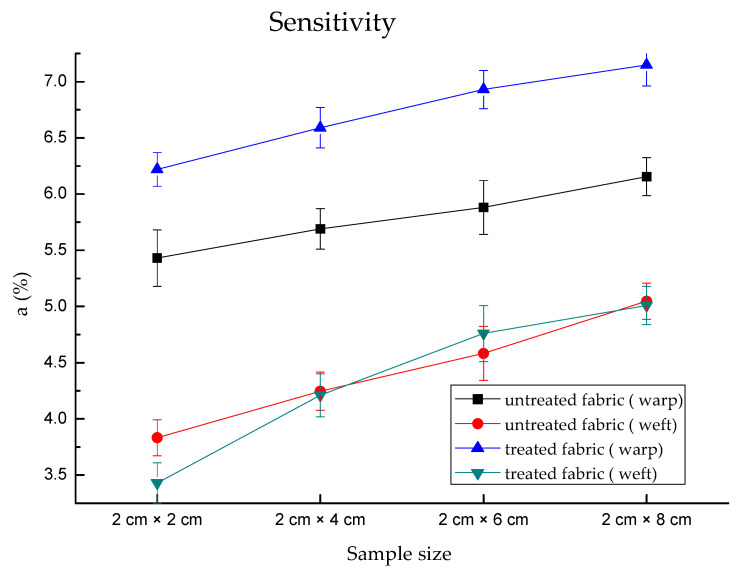
The sensitivity of untreated and treated fabrics. Treated fabric in the warp direction had higher sensitivity.

**Figure 11 polymers-13-01015-f011:**
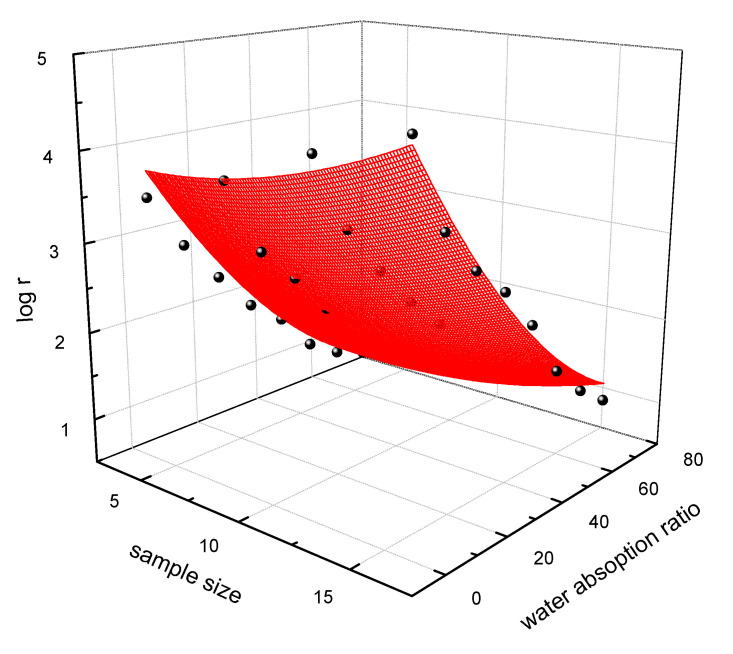
The plot of the predicted mode of electrical resistance.

**Table 1 polymers-13-01015-t001:** The knitted fabric parameter (treated fabric had higher mass, lower thickness, and higher loop density due to the shrinkage of finishing).

Mass (g/m^2^)	Thickness (mm)	Structure	Wales per 5 cm	Courses per 5 cm	Finishing
205	1.84	1 + 1 rib	26	36	Non-finishing
273	1.4	1 + 1 rib	30	44	Hydrophilic finishing

**Table 2 polymers-13-01015-t002:** The *p* values of significances among samples 2 × 2, 2 × 4, 2 × 6 and 2 × 8 under different water absorption ratio.

Directions	0%	10%	20%	30%	40%
Warp	0.000	0.000	0.000	0.002	0.002
Weft	0.000	0.000	0.000	0.000	0.000

**Table 3 polymers-13-01015-t003:** The *p* values of significances among samples 2 × 2, 2 × 4, 2 × 6 and 2 × 8 under different water absorption ratio.

Directions	0%	10%	20%	30%	40%	50%	60%	70%
Warp	0.000	0.000	0.002	0.000	0.000	0.000	0.000	0.000
Weft	0.000	0.000	0.000	0.000	0.000	0.000	0.000	0.000

## Data Availability

The data presented in this study are available on request from the corresponding author.

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
