# Peer review of "Electrical Resistance of Stainless Steel/Polyester Blended Knitted Fabrics for Application to Measure Sweat Quantity"

_polymers, 2021, doi:10.3390/polym13071015_

Round 1
Reviewer 1 Report
Comments to the Authors of paper „Electrical resistance of stainless steel/polyester blended knitted fabrics for application to measure sweat quantity”
There are several remarks to the Authors and in my opinion they should somehow answer to them in their paper or at least somehow make a comment on them.
- Observed changes of resistivity results from addition drops of NaCl solution - are these measurements reproducible ?
- What will happen after textile washing ? Again are the observed resistivity changes reproducible ? There is mentioned unstability of ink-printed resistors (page 2 lines 81 – 83). How it is in this case ?
- Stainless steel wires after numerosus bendings have tendency to break and form sharp endings – can it be harmfull to the persons using such a textiles ?
- There are too many easy to correct but irritiating editorial errors:
Page 1 lines 5;6;7 - missing addresses
Page 3 line 105 - what is DEW agent ?
Line 106 - should be 15 min (missing space)
Line 107 - should be Table 1
Line 108 - should be 3 µl
Line 115 - should be The knitted fabrics…
Line 118 – should be The contact angle…
Line 120 – should be 2 cm x 2 cm; 2 cm x 4 cm….
Line 123 – how many drops were deposited ? Always the same number ?
Line 125 – how is defined log r ?
Lines 136 and 141 – should be 2 cm; 4 cm; 6 cm; 8 cm
Line 149 – should be The electrical…
Line 183 – should be respectively (figure…
Line 206 – should be 2 cm
Line 231 - should be 2 cm
Line 269 – should be The comparison…
Line 286 – should be The sensitivity…
Line 297 - should be The plot….
- In the paper with numerical data there should be given scale (numerical values of results) and bench mark – reference point. In this paper there is no reference point, we do not know if the results are interesting, good, fantastic or just the minor ones. Reference point can be hidden, without name of the producer but should be given

Author Response
Dear reviewers,
Thank you for your comments. The manuscript was revised.
Best regards
Qing Chen
Reviewer 2 Report
The work presents alternative materials for the detection of Skin wetness and sweat rate. The authors use polymer/metal blends produced by a relatively simple methodology for electrical testing, simulating smart textiles.
- The Introduction presents relevant content but the writing needs to be revised. As an example, in the Introduction section, the phrase "A moisture monitoring system with textile-integrated sensors for wound healing assessment [21]." it needs to be rewritten to make sense.
- The authors do not present the state of the art of the research subject. Note that of the 27 citations, apparently only 4 of them are from the last 5 years. The importance of the research theme and the proposal presented in the work in relation to what is available in the literature cannot be analyzed in this way.
- In the description of Fig. 1, "Stainless steel fiber and polyester blended conductive yarn." Please detect each material.
- "Figure 2 is a fabric schematic diagram". Fig. 2 is not a schematic diagram!
- The writing of section 2 needs to be vastly improved. Phrases and verb tenses need to be revised.
- Sections 2 and 3 can be combined with the title "2. Materials and methods" with 2.1 Materials and 2.2 Methods.
- In section 2, "The treated fabrics had 33% higher mass and 40% lower thickness that untreated fabrics. This is because the fabric shrinkage happened during wetted finishing process. As a result, the treated fabric had more loop numbers in unit area "This is an experimental result! It is not a description of the material.
- "Figures 4 (a) and (b) plot log (electrical resistance) values ​​as a function of water absorption for untreated fabrics." Why do untreated samples have low R2 for the suggested functions? What is the physical meaning of this? The authors only present results without discussing them.
- On several occasions, the measurement units are forgotten.
- The authors only describe the sample dimensions in the abstract, before presenting the results for each of them in section 4.
- Furthermore, what is the physical relevance of these experimental dimensions for later results with real dimensions?
- As the authors are dealing with material said to be alternative to proposed applications, a statistical analysis is necessary in order to verify significant differences in results presented in Figs. 5 and 6. How many independent samples were tested for each different sample size, so that the results were considered for publication?
- Are there significant differences in the results presented in section 4.3 for the different samples? The results presented do not present significant differences confirmed by any method of analysis. Even considering different directions, this difference may not exist.
- The descriptions in the Figures are very vague and do not contain the appropriate information for the reader.
Author Response

(The authors gave the same response as above.)
